# Adsorbate-induced lattice deformation in IRMOF-74 series

Sudi Jawahery[1], Cory M. Simon[1], Efrem Braun[1], Matthew Witman[1], Davide Tiana[2], Bess Vlaisavljevich[3] & Berend Smit[1,2]

IRMOF-74 analogues are among the most widely studied metal-organic frameworks (MOFs) for adsorption applications because of their one-dimensional channels and high metal density. Most studies involving the IRMOF-74 series assume that the crystal lattice is rigid. This assumption guides the interpretation of experimental data, as changes in the crystal symmetry have so far been ignored as a possibility in the literature. Here, we report a deformation pattern, induced by the adsorption of argon, for IRMOF-74-V. This work has two main implications. First, we use molecular simulations to demonstrate that the IRMOF-74 series undergoes a deformation that is similar to the mechanism behind breathing MOFs, but is unique because the deformation pattern extends beyond a single unit cell of the original structure. Second, we provide an alternative interpretation of experimental small-angle X-ray scattering profiles of these systems, which changes how we view the fundamentals of adsorption in this MOF series.

[1] Department of Chemical and Biomolecular Engineering, University of California, Berkeley, California 94720, USA. [2] Laboratory of Molecular Simulation, Institut des Sciences et Ingénierie Chimiques, École Polytechnique Fédérale de Lausanne (EPFL), Rue de l'Industrie 17, CH-1951 Sion, Switzerland. [3] Department of Chemistry, Northwestern University, Evanston, Illinois 60208, USA. Correspondence and requests for materials should be addressed to B.S. (email: berend.smit@epfl.ch).

Metal-organic frameworks (MOFs) are a class of porous crystals that hold promise for adsorption-based applications, such as carbon capture[1], hydrogen[2] and methane storage[3] and noble gas separations[4]. MOFs can be tailored for a given application by tuning the size, shape and chemistry of their constituent building blocks[5]. Effective optimization of MOFs for adsorption-based applications depends on our ability to engineer their chemistry to modulate their selectivity and capacity for different gas molecules. A fundamental understanding of the mechanisms of adsorption in MOFs and their response to gas adsorption is therefore crucial.

Several MOFs whose crystal structures respond to adsorbing gas molecules by undergoing reversible, structural transitions have been reported in the literature[6] and studied theoretically[7–9]. Adsorbate-induced deformations of MOFs can cause dramatic changes in unit cell parameters, as in the cases of breathing or swelling MOFs. Swelling mechanisms that allow MOFs to expand as pores fill with adsorbates have been identified[10]. More complex breathing mechanisms have also been observed where, upon insertion of some adsorbates at low gas phase pressures, the MOF exhibits a transition from an expanded, porous state to a more collapsed state, before expanding yet again to the porous state at higher gas phase pressures[11–13]. The structural transitions can also be more subtle, as in the case of MOFs whose ligands undergo a rotation upon gas adsorption to accommodate more molecules yet maintain an approximately rigid unit cell[14]. Recently, a rare negative gas adsorption phenomenon was observed in a MOF and directly connected to the cooperative motion of many atoms across a large unit cell[15]. The crystal structure of a MOF at a given set of conditions is determined by trade-offs between intrahost, host-adsorbate, and adsorbate–adsorbate energies, as well as the mechanical stress induced by the pressure of the gas[16]. Deformation of the framework raises its energy in the absence of gas, but is often compensated for by more favourable host–adsorbate and/or adsorbate–adsorbate interactions.

Adsorption-induced deformation is a well-documented example of how adsorption can deviate from what is expected in perfect, rigid crystals. By contrast, adsorption in rigid lattices is known to be energetically driven only by host–adsorbate and adsorbate–adsorbate interactions. Intuitively, these interactions should be of a length scale associated with the adsorbate molecule. For example, water adsorbates may feel interactions on a long length scale because of water's unique hydrogen-bonding pattern[17], while noble gas adsorbates, which interact primarily through dispersion forces, feel no such effect and the length scale of their interactions is typically limited to the scale of their atomic radius. Within MOFs, unique adsorption mechanisms such as cooperative phenomena have previously been identified along with the system-specific interactions that drive them[18].

In this context, recent work by Cho et al.[19] stands out in suggesting that adsorbate-adsorbate interactions across channel walls can drive adsorption in MOFs. Cho et al. studied the behaviour of different adsorbates in the IRMOF-74 series and presented in situ small-angle X-ray scattering (SAXS) data. The data for argon adsorption show evidence for the formation of extra argon adsorption domains in IRMOF-74-V-hex. The proposed extra argon adsorption domains take the form of a periodic superlattice, the dimensions of which extend beyond the size of a single channel (see Fig. 1, magenta lines). The four corners of a proposed superlattice unit cell represent channels with high argon density relative to the six adjacent channels. Cho et al. explain the superlattice by proposing a new mechanism in which cross-channel interactions of argon at 87 K stabilize the extra adsorption domains. Cho et al. further speculate that this is induced by a contraction of the unit cell.

The work by Cho et al. is the first report of an adsorption mechanism that involves cross-channel adsorbate interactions in a MOF. We therefore found it useful to use molecular simulation techniques to further quantify the importance of these cross-channel argon interactions. Unlike the experimental system, we can tune interactions in our simulations to directly compare a system in which argon–argon interactions are restricted to act only within a single channel with a system in which cross-channel interactions are allowed. Our simulations of argon adsorption in IRMOF-74-V, a MOF composed of Mg$^{2+}$ atoms and expanded dioxidoterephthalate ligands that include five aromatic rings[20], show, however, that cross-channel argon interactions have only a minor effect on the adsorption isotherm and do not lead to the argon superlattice described by Cho et al. We did discover that argon adsorption induces a deformation of the crystal lattice, and we demonstrate that this deformation can account for the signatures observed in the X-ray pattern measured by Cho et al.

## Results

**Adsorption isotherms.** We used Grand-canonical Monte Carlo (GCMC) simulations to compute argon adsorption isotherms in IRMOF-74-V. In Fig. 2a we compare our isotherms with the isotherm measured by Cho et al. The snapshots in Fig. 2b illustrate that argon adsorbs in layers on channel walls: at 0.001 bar, there is a single layer of argon atoms in the channel, followed by a second layer at 0.10 bar. The pore completely fills a little beyond 0.40 bar. A comparison of the snapshots in Fig. 2b and the isotherm in Fig. 2a shows that the jump in the isotherm occurs exactly before the pores flood with argon. The jump in the isotherms indicates capillary condensation in the pores[21], which have an approximate diameter of 40 Å. The hysteresis that we observe upon desorption is further evidence for capillary condensation and is a well known phenomenon in structures containing mesopores[22]. Our simulations are in good qualitative agreement with the experimentally measured isotherm, capturing the inflection point in argon uptake in the range of 35–45 kPa. Cho et al. also reported hysteresis in their experimental isotherm but over a smaller pressure window. The large simulated Henry coefficient, apparent in the overestimation of loading at low

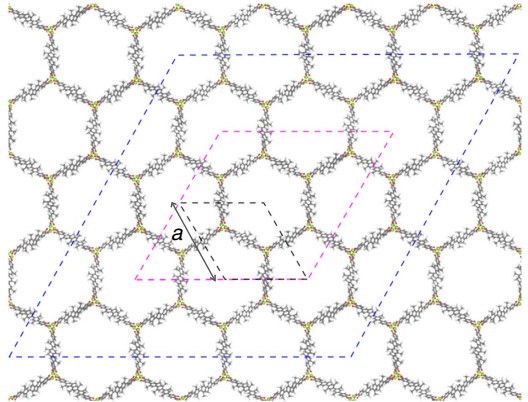

**Figure 1 | Different unit cells of the IRMOF-74-V lattice that are relevant to this study.** The primitive unit cell used for DFT structure relaxation is marked in black, the simulation box used in NPT simulations is marked in blue and the superlattice dimension proposed by Cho et al. is marked in magenta. Cho et al. propose that the corners of the superlattice represent regions of high adsorbate density. Unit cell parameter $a$ is marked in black next to the primitive unit cell. Magnesium, oxygen, carbon and hydrogen atoms are marked in lime green, red, grey and white, respectively.

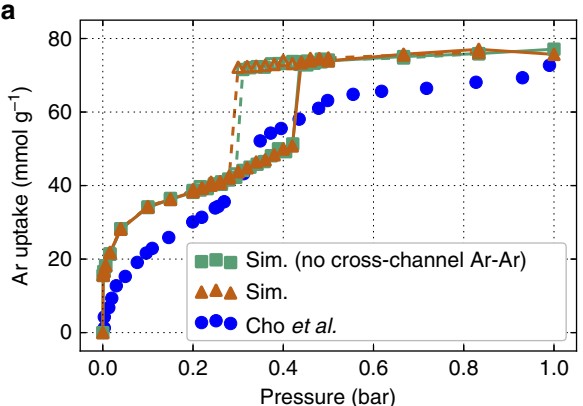

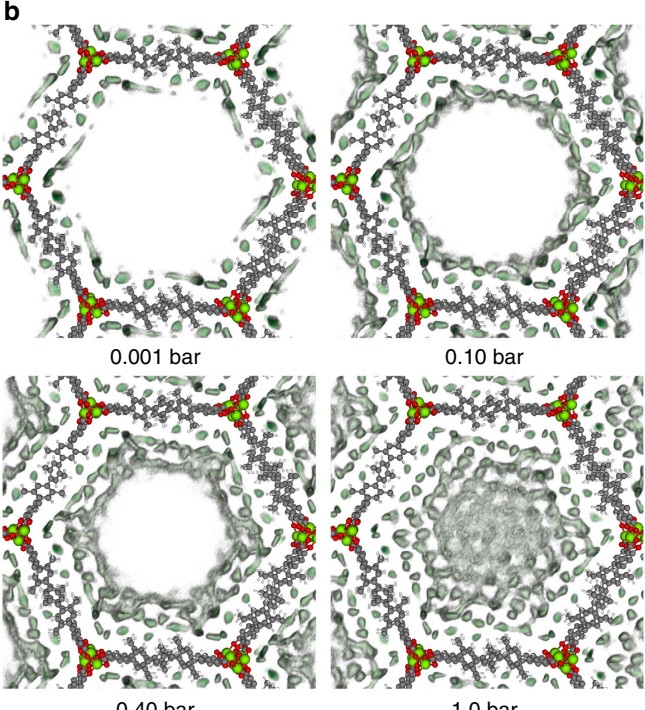

**Figure 2 | Characteristics of argon adsorption in a rigid lattice model.**
(**a**,**b**) Simulated and experimental argon adsorption isotherms in IRMOF-74-V at 87 K are shown in **a** and snapshots of argon loading at select pressures are shown in **b**. In the snapshots, argon density is coloured in dark green and the rigid lattice model is also shown. The orange curve in **a** is a simulated argon isotherm. The teal curve in **a** is a simulated argon isotherm where argon–argon interactions across channels are artificially turned off; that is, argon adsorbates interact with each other within the same hexagonal channel, but two argon atoms in different channels do not interact. Adsorption and desorption portions of the simulated isotherms are marked with closed and open markers, respectively. The blue points in **a** are the experimental isotherm measured by Cho *et al.*

pressures, can be attributed to inaccuracies in the force field description of noble gas–MOF interactions, and is in similar agreement compared with other studies on noble gas adsorption in MOFs[23]. At intermediate and high loadings, the quantitative differences between the experimental and simulated isotherms may be related to the fact that we use a rigid lattice model for all simulations and assume a perfect crystal structure, which is experimentally impossible to synthesize.

The shape of the argon adsorption isotherm was presented by Cho *et al.* as a piece of evidence in support of cross-channel

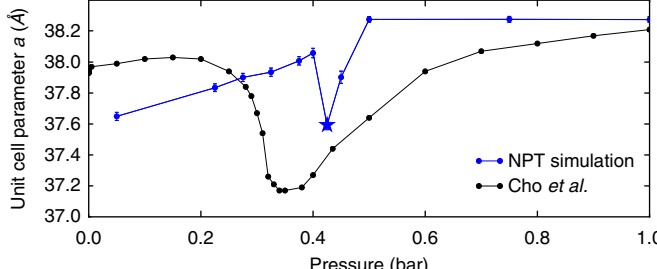

**Figure 3 | Unit cell parameter *a* of IRMOF-74-V as a function of pressure.**
Unit cell parameter *a* is denoted as a double-headed black arrow next to the primitive unit cell in Fig. 1. NPT simulation data is shown in blue and the experimental data of Cho *et al.* is shown in black. Simulation data points are averages and error bars show the standard deviation. The loadings imposed in the NPT simulation are commensurate with the simulated standard lattice adsorption at that pressure. Stars represent conditions at which lattice deformation was observed, as shown in Fig. 4.

interaction induced argon ordering as the inflection point in the isotherm was attributed to the formation of extra adsorption domains. To study the effect of cross-channel interactions we compared GCMC simulations of a system in which argon can interact across channels with a system in which we (artificially) restricted argon–argon interactions to contribute only if the argon atoms are in the same channel. The adsorption mechanism proposed by Cho *et al.* suggests that by artificially turning off interactions between argon atoms that are not in the same channel, we should obtain a qualitatively different adsorption behaviour. A comparison of the two simulated isotherms in Fig. 2a reveals that cross-channel argon–argon interactions have little to no effect on the isotherm. Argon atoms in different channels are too far apart to interact significantly. We quantify this effect in Supplementary Fig. 4.

**Lattice deformation**. An important difference between our GCMC simulations for which isotherms are presented in Fig. 2a and experiments is that we assume a rigid crystal structure, while experimental data shows that the unit cell parameter *a*, shown in Fig. 1, does change upon adsorption. To study the effect of argon adsorption on the crystal structure of IRMOF-74-V, we use flexible lattice simulations. In these simulations, we use for each channel a fixed loading given by the pressure of the simulated isotherms (Fig. 2a) and monitor the change of the IRMOF-74-V structure in terms of the lattice parameter as a function of pressure. Figure 3 shows that our simulations agree well with the experimentally observed behaviour in the lattice parameter. Similar to the experiments, we observe at low pressures that the crystal swells when the pressure and loading are increased. At intermediate pressures, however, we see a dip in the lattice parameter, while at high pressures the crystal swells again. Upon further inspection of the lattice at 0.425 bar, we discover that the unit cell parameter *a* does not only shrink but the lattice also deforms (Fig. 4) such that it is no longer represented by the crystal structure shown in Fig. 1.

The standard lattice of the IRMOF-74 series is composed of channels in the shape of regular hexagons. Four distinct types of channels exist in the deformed lattice and are shown in Fig. 4. The central Channel 1 resembles a channel in the standard lattice. Channels 2, 3 and 4 are irregular hexagons that have a smaller volume than Channel 1 and form a spiral shape around the central, regular channel. Two irregular channels of the same type are always found directly across the regular channel from each other. This pattern imposes a geometric constraint that causes the observed deformation to be coherent and crystalline. If one

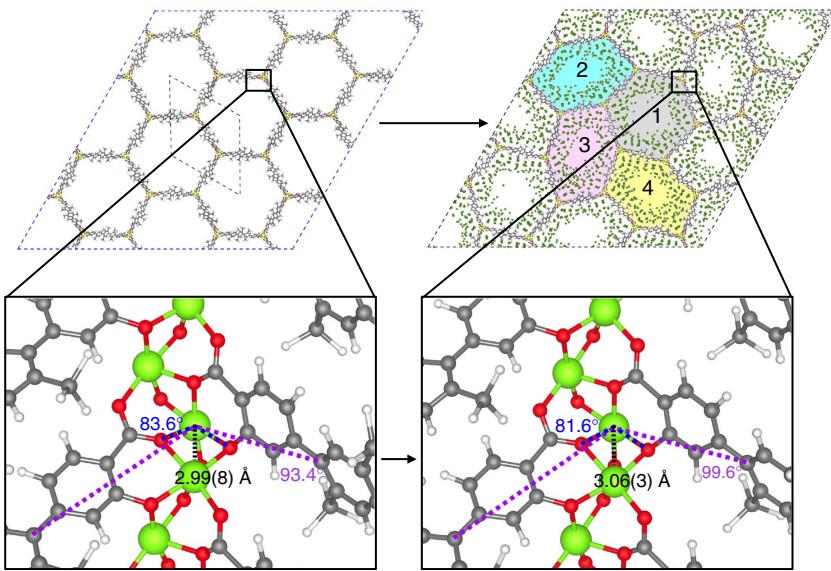

**Figure 4 | Representations of lattice deformation.** Black dotted lines represent the unit cell of standard and deformed crystal lattices. The deformed lattice shows Channel 1 in grey, Channel 2 in blue, Channel 3 in pink and Channel 4 in yellow. The views of a standard right-handed metal helix from inside a standard and irregular channel are also shown. Magnesium, oxygen, carbon, hydrogen and argon atoms are marked in lime green, red, grey, white and dark green, respectively. The distance between metal atoms is larger in the deformed helix, indicating that the helix becomes wider. Angles between atoms on neighboring linkers change accordingly.

channel deforms, the lattice must deform as well to prevent massive strain.

The observed deformation is associated with a subtle change in geometry of some metal helices. Figure 4 shows two views of right-handed metal helices from the interior of a regular channel in a standard lattice and an irregular channel (Channels 2, 3 or 4) in a deformed lattice. Metal helices are deformed if they join Channel 1 hexagons with two irregular channels. The deformed metal helix is slightly wider than the standard metal helix, as shown by the larger distance between metals. To maintain an appropriately square pyramidal coordination geometry, metal-bound oxygen atoms shift their positions. This rearrangement of atoms is propagated along the flexible linkers and causes a change in the vertex angle of the irregular channels.

In the GCMC simulations that we have already presented, we have used only a rigid standard IRMOF-74-V lattice. It is therefore important to determine whether a rigid deformed lattice can lead to a qualitatively different isotherm. To accomplish this we investigated the adsorption properties of Channels 1, 2, 3 and 4 in the deformed lattice separately. Figure 5a shows that while the argon uptake in all four channels is similar at low pressures, Channels 2, 3 and 4 all adsorb less argon than Channel 1 at high pressures. The difference in loading at high pressures is easily rationalized by the difference in volume between channels.

The individual channel isotherms demonstrate that Channels 2, 3 and 4 all experience a jump in loading at a lower pressure than Channel 1. The difference in pore filling pressures across the channels is in agreement with the expectation that capillary condensation occurs at lower pressures in smaller channels[24]. Therefore, in the very narrow pressure window between 0.40 and 0.425 bar, the smaller deformed channels actually adsorb a higher density of argon than Channel 1. A comparison of Fig. 5 with Fig. 3 shows that this narrow pressure window corresponds with the pressure at which the deformed lattice was observed. The composite deformed lattice isotherm in Fig. 5b was made by averaging together the four individual channel isotherms shown in Fig. 5a. The effect of lattice deformation can be seen by comparing the deformed composite isotherm to the Channel 1 isotherm, which resembles a channel in a standard lattice.

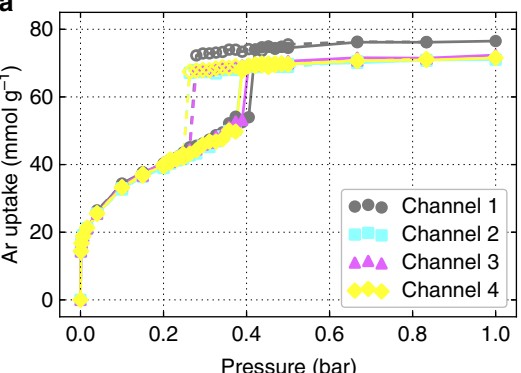

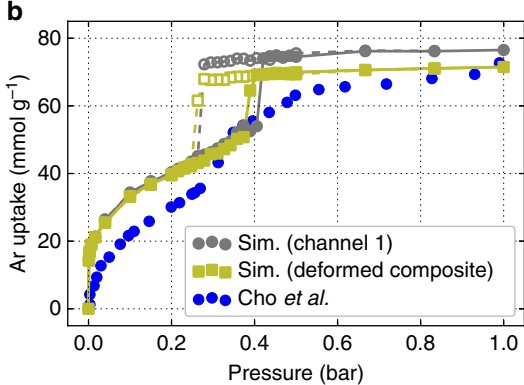

**Figure 5 | Argon adsorption isotherms in the deformed lattice.**
(**a,b**) Adsorption isotherms for each of the four channels of the deformed lattice, as shown in Fig. 4, are shown in **a** and match the colour code shown in Fig. 4. The yellow curve in **b** is the deformed lattice composite isotherm obtained by averaging together the isotherms for the four different channels. Adsorption and desorption portions of the simulated isotherms are marked with closed and open markers, respectively. The blue points in **b** are the experimental isotherm measured by Cho *et al.*

The Channel 1 isotherm shown in Fig. 5 and rigid lattice isotherm shown in Fig. 2a are different. The rigid lattice used in Fig. 2a is the density functional theory (DFT)-optimized structure, whereas the deformed lattice and individual channel structures result from molecular simulations and are governed by the force field; consequently, the two structures vary and have slightly different sizes and linker orientations. For this reason, Channel 1 should be compared with the deformed channels and lattice rather than the DFT-optimized structure. In the pressure range where the deformed lattice was observed (near 0.425 bar), the effect of lattice deformation is to cause a more smooth increase in loading as each of the channels will fill at a slightly different pressure in the deformed lattice.

**X-ray spectra.** In our simulations, we find no evidence to support the cross-channel interaction mechanism or adsorbate super-lattice proposed by Cho *et al.* At this point, it is important to note that Cho *et al.* assumed that for all pressures, the IRMOF-74 lattice maintains its crystal structure and does not deform. Indeed, it was reasonable to make this assumption because several previous studies of IRMOF-74 have suggested that the metal-oxygen nodes are relatively rigid[25,26]. It is therefore important to investigate the impact of the observed deformation on the X-ray pattern of the system. We compare the X-ray signature associated with our deformed lattice with the experimental X-ray data. The simulated deformed X-ray pattern shown in Fig. 6a is intriguingly similar to the experimental SAXS profiles. Cho *et al.* observed the

appearance of peaks at $q = 0.25\,\text{Å}^{-1}$ and $q = 0.42\,\text{Å}^{-1}$ during argon adsorption in one of the IRMOF-74 analogues studied, IRMOF-74-V-hex, at intermediate pressure and loading conditions. Cho *et al.* also observed broad peaks at $q = 0.10\,\text{Å}^{-1}$ for all IRMOF-74 analogues studied. Experimentally, peaks associated with the standard lattice did not disappear in any of the IRMOF-74 analogues studied, a fact that was specifically interpreted by Cho *et al.* to mean that the lattice was not deforming[19]. As we show in Fig. 6a, deformation of the IRMOF-74-V lattice produces new X-ray peaks at all of the $q$-values that were experimentally noted without causing the standard lattice peaks to disappear. Although Cho *et al.* attributed the additional peaks in their SAXS profiles to long-range ordering of argon atoms, we believe that framework deformation is a more likely physical origin of this signal.

As is observed in the experimental data, the standard lattice peaks are preserved in the deformed lattice X-ray pattern, but not because the lattice maintains its structure. The standard lattice of IRMOF-74-V exhbits peaks in the X-ray pattern at $q$-values near 0.20 and $0.33\,\text{Å}^{-1}$, marked in red and blue in Fig. 6a. These peaks are associated with two independent crystallographic spacings, the $hk = 10$ and 11 Miller face spacings, marked in red and blue respectively in Fig. 6c. The deformed lattice X-ray pattern exhibits peaks of similar magnitude in similar positions as the standard lattice, also marked in red and blue in Fig. 6a. The standard peaks appear in the deformed X-ray pattern because the deformed lattice includes crystallographic spacings that are geometrically similar to those in the standard lattice, the

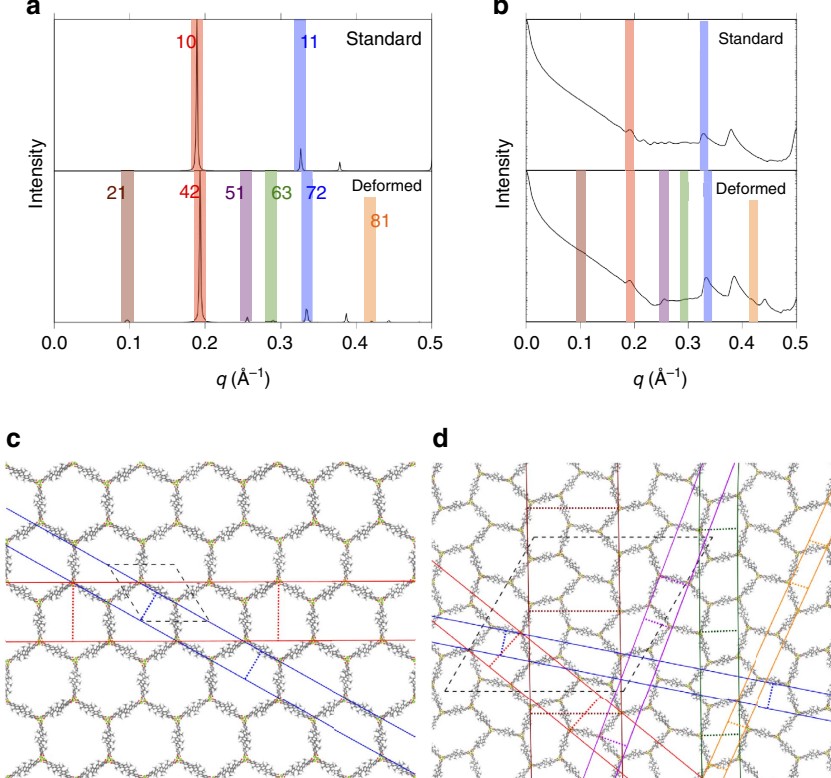

**Figure 6 | Simulated X-ray spectra and associated standard and deformed IRMOF-74-V lattices. (a,b)** X-ray patterns for the MOF are shown in **a** and SAXS profiles for the argon are shown in **b**. In the X-ray patterns, the standard lattice corresponds to vacuum and the deformed lattice to a pressure of 0.425 bar and loading commensurate with the adsorption isotherm. In the SAXS profiles, the deformed lattice and loading correspond to a pressure of 0.425 bar, but a standard lattice is artificially given a loading corresponding to 0.425 bar. **(c,d)** Peaks in the patterns are marked with a colour-code that corresponds to the crystallographic spacings shown in **c,d**, which are labelled snaphots of standard (**c**) and deformed (**d**) lattices. Red and blue lines in **c** represent $hk = 10$ and 11 Miller faces, respectively. The red, blue, orange, green, purple and maroon lines in **d** represent the $hk = 42$, 72, 81, 63, 51, and 21 Miller faces, respectively.

$hk = 42$ and 72 Miller face spacings shown in red and blue in Fig. 6d. The peaks in the deformed X-ray pattern that are related to the standard peaks are shifted slightly to the right. This is a consquence of the overall decrease in unit cell parameter. The apparently larger shift associated with the $hk = 11$ compared with the $hk = 10$ Miller face spacing does not indicate a larger change in interplanar spacing. Rather, the $q$-value at which an interplanar spacing appears is more sensitive at smaller spacings ($q = 2\pi/d$, where $d$ is interplanar spacing).

New peaks observed by Cho *et al.* can be explained by the increased number of crystallographic spacings in the deformed lattice. The peak observed experimentally at $q = 0.25\,\text{Å}^{-1}$ corresponds to the $hk = 51$ Miller face spacing in the deformed lattice, marked in purple in Fig. 6d. The peak observed experimentally at $q = 0.42\,\text{Å}^{-1}$ corresponds to the $hk = 81$ Miller face spacings in the deformed lattice, marked in orange in Fig. 6d. These two peaks are related to expansion and compression of the $hk = 11$ Miller face spacing of the standard lattice, respectively. The broad peak experimentally observed at $q = 0.10\,\text{Å}^{-1}$ appears at the same $q$-value as the peak associated with the $hk = 21$ Miller face spacing in the deformed lattice, marked in maroon in Fig. 6d. The $hk = 21$ Miller face spacing in the deformed lattice is unique because it spans the length of more than one hexagon.

Simulated SAXS profiles that account only for argon light scattering are shown in Fig. 6b. This figure demonstrates that argon does not organize differently across channels as a result of deformation. In both cases, peaks appear in the SAXS profiles at $q$-values that correspond to peaks in the powder X-ray diffraction pattern of the respective lattice. This feature is due to the fact that argon is adsorbing on the walls of the channel, as shown in Fig. 2b, and the SAXS profile peaks are a reflection of the bounding lattice.

There are some discrepancies between the X-ray peaks shown in Fig. 6a and the SAXS profile measured by Cho *et al.*, but these differences can be explained. The standard peaks in the simulated X-ray pattern associated with the $hk = 10$ and $hk = 11$ Miller face spacings do not shift as noticeably in the work by Cho *et al.* We attribute this to finite-size effects in our simulation, as the necessarily small size of our simulation superlattice (4 channels × 4 channels) imposes a uniform deformation on opposite sides of the simulation box via periodic boundary conditions. A larger superlattice approaching the size of the experimentally realized system would experience less perfect and perhaps less widespread deformation, reducing the overall effect on interplanar spacing. We present data for a slightly larger simulation system in Supplementary Figs 6 and 7, and find the absence of a peak at $q = 0.10\,\text{Å}^{-1}$ indicative of less perfect deformation. In addition, the small peak we observe at $q = 0.28\,\text{Å}^{-1}$, associated with the $hk = 63$ Miller face spacing in the deformed lattice, is not noted by Cho *et al.*, but we believe this peak may be indistinguishable from the much more intense standard peak near $q = 0.33\,\text{Å}^{-1}$ in the experimental SAXS data[19].

## Discussion
Using classical simulations, we predicted that the IRMOF-74 series can undergo an adsorbate-induced deformation. The deformed framework reproduces key features that were experimentally observed during argon adsorption by Cho *et al.*, namely changes in the lattice parameter and X-ray pattern. We therefore propose the deformation as an alternative hypothesis to a superlattice of different argon densities in neighboring channels. As both interpretations can explain the additional X-ray peaks, it would therefore be interesting to see whether additional experiments can confirm our proposed deformation.

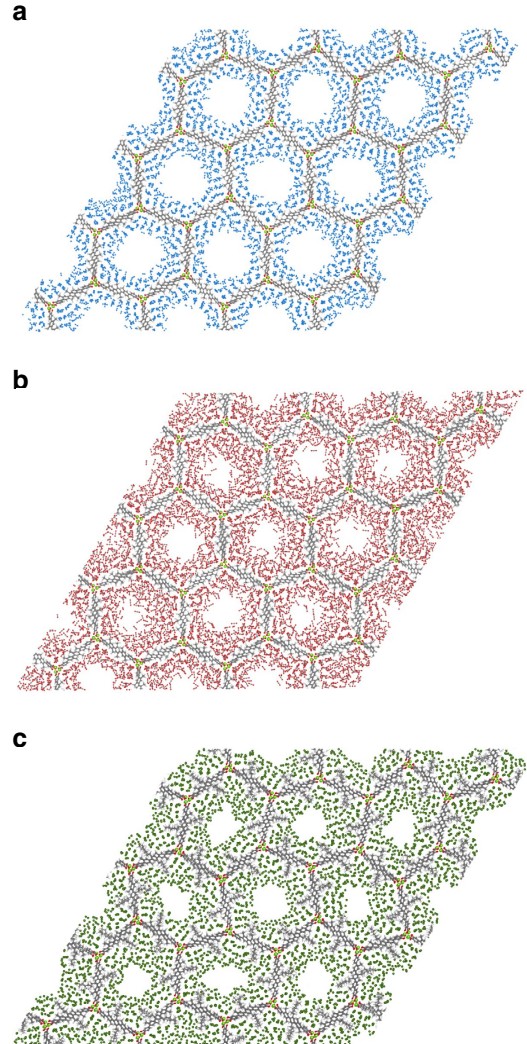

**Figure 7 | Snapshots of deformed lattices of other adsorbate–framework systems.** IRMOF-74-V lattices loaded with $N_2$ and $CO_2$ are shown in **a** and **b**, respectively, and an IRMOF-74-V-hex lattice loaded with argon is shown in **c**. We accessed deformation in these systems by imposing the minimum unit cell parameter $a$ reported by Cho *et al.* during the adsorption process, and by loading the lattice with the commensurate number of adsorbate molecules. The systems were then simulated in the NVT ensemble at temperatures corresponding to the measurements by Cho *et al.* (77 K for $N_2$, 194 K for $CO_2$ and 87 K for argon). Nitrogen, carbon, oxygen and argon atoms are marked in blue, grey, red and dark green, respectively.

The appearance at intermediate pressures and disappearance at high pressures of adsorbate-induced deformation provides us with insight into how the deformation of IRMOF-74-V is stabilized. By comparing Fig. 3 with the adsorption isotherms, we note that the deformation is observed before the channels are filled and disappears as the lattice saturates with argon. As the channels fill, adsorbate-adsorbate forces in the dense argon environment destabilize the deformation and force the lattice to return to its standard size and structure, potentially to accommodate more adsorbates. Although we do not characterize IRMOF-74-V as a breathing MOF, this observation allows us to draw a parallel between this system and breathing MOFs, which collapse at intermediate loadings to improve adsorbate interactions and expand at high pressures to accomodate more adsorbates[11].

GCMC simulations showed no evidence of the argon adsorbate superlattice in a rigid framework as proposed by Cho *et al.* However, GCMC simulations did demonstrate that, in a very narrow pressure range, irregular hexagonal channels in the deformed lattice can adsorb more argon compared with regular hexagonal channels. This implies the opposite conclusion of Cho *et al.*, who proposed small regions of high loading separated by large regions of low loading. Our GCMC simulations of individual hexagonal channels suggest that regions of low loading can be surrounded and separated in space by regions of high loading.

Our simulations show that the observed breathing effect is a subtle balance between the energetics of the deformation of the lattice and interactions of argon atoms. For example, Cho *et al.* observed that a dip in the lattice parameter occurs over a very narrow pressure range in IRMOF-74-V-hex and over a less narrow pressure range in IRMOF-74-V. A possible explanation is that the presence of the side group in IRMOF-74-V-hex makes it more difficult to deform the lattice. However, further work is needed in refining our model to reliably study the effects of these subtle differences.

The arguments we have presented suggest that the observed deformation is not limited to argon adsorbing in IRMOF-74-V, but can also occur in other adsorbate-framework systems within the IRMOF-74 series. In Fig. 7, we show that the same deformation indeed occurs with other adsorbates and IRMOF-74 analogues. Figure 7 demonstrates that the adsorbate identity affects the extent of deformation, which we describe in more detail in Supplementary Figs 8 and 9. We also modelled argon adsorbing in IRMOF-74-III and IRMOF-74-VII, but, as we show in Supplementary Fig. 10, we did not access deformation in these structures. This suggests a pore size dependence, wherein argon stabilizes deformation more in pores that are on the scale of 40 Å compared with pores that are 15 Å smaller or larger.

The appearance and disappearance of the observed deformation is similar to the phase transitions of a conventional breathing MOF, where the flexible unit cell is open in vacuum, closes to become narrow upon gas adsorption, then opens again as the pores saturate. The deformation we observe, however, is unique because it cannot be described with a single unit cell of the original structure. Unlike conventional breathing MOFs in which neighboring pores simultaneously close at intermediate pressures and open at high pressure, neighboring pores in the IRMOF-74 series necessarily deform differently and form a repeating, crystalline pattern. The length scale required to study this deformation is therefore much larger than a single unit cell of the material.

## Methods

**DFT structure relaxation.** All DFT calculations were performed with VASP[27] version 5.3.5, using projector augmented wave method potentials[28–30]. We used a revised version of the van der Waals dispersion-corrected density functional (rev-vdW-DF2)[31] that was implemented in VASP by Klimeš *et al.*[32] using the algorithm of Román-Pérez *et al.*[33].

We relaxed the experimental crystal structure via DFT calculations using a procedure that has been shown to lead to reasonable lattice constants and bond lengths in IRMOF-74 (ref. 34). The resulting relaxed unit cell was a starting point for all MC and molecular dynamics (MD) simulations. Periodic DFT optimizations of the IRMOF-74-V and IRMOF-74-V-hex structures were done starting from respective 228-atom and 318-atom triclinic primitive unit cells obtained after removing solvent and reducing the symmetry of the experimental structure[20]. Integration over the Brillouin zone was carried out using Γ-point sampling. The plane-wave basis set was cutoff at 600 eV, and the wave function energy convergence criterion was set to $10^{-5}$ eV. The atomic positions were optimized until all forces were smaller than 0.05 eV Å$^{-1}$. Point charges were then assigned to the framework atoms using the REPEAT scheme[35] with the electrostatic potential generated from the structural relaxation.

**Molecular model.** We use a flexible framework model based on the work of Greathouse and Allendorf[36], which demonstrated that modelling metal-linker bonds classically as non-bonded interactions gives an accurate description of the flexibility of MOFs. The use of non-bonded interactions to model metal-linker bonds allows us to study changes in the geometry at the coordination centers of IRMOF-74-V. More details about the framework model are available in Supplementary Fig. 1, Supplementary Table 1 and Supplementary Methods.

In our model for IRMOF-74-V, bond, angle, dihedral and torsion parameters for linker molecules were taken from the consistent valence force field (CVFF)[37]. Metals are modelled using a cationic dummy model parameterized by Duarte *et al.*[38] to delocalize charge and specifically model $M^{2+}$ ions in an octahedral coordination environment. Lennard-Jones parameters taken from CVFF for linker atoms and from Duarte *et al.* for metal atoms were used with geometric mixing rules to compute non-bonded framework–framework interactions. Implementation of this force field and unit cell relaxation led to good agreement with the DFT energy-minimized structure (less than 1% difference of any unit cell parameter, shown in Supplementary Table 2). Supplementary Figure 2 shows reasonable agreement of the simulated infrared (IR) spectrum of IRMOF-74 using our molecular model with experiment[39].

For our models of argon–argon and argon–MOF interactions, Lennard-Jones parameters for argon were taken from work by Brown and Clarke[40]. Non-bonded interactions between argon, linker and metal atoms were calculated by using geometric mixing rules with Lennard-Jones parameters from Brown and Clarke[40], CVFF[37] and Duarte *et al.*[38], respectively, as was similarly done by Macedonia *et al.* to calculate interactions of argon with zeolites[41]. Lennard-Jones interactions and short-range (real space) coulombic interactions were truncated at 10 Å, and Ewald summations were used to calculate long-range coulombic interactions.

**NVT and NPT molecular dynamics simulations.** Changes in the lattice of IRMOF-74-V as a function of loading were simulated using MD simulations in the canonical (NVT) and isobaric–isothermal (NPT) ensembles using the LAMMPS molecular software package[42]. Using the observations of Cho *et al.* as a guiding reference, we first slowly decreased the unit cell parameter *a* of IRMOF-74-V loaded with argon by 90% while simulating the lattice in the NVT ensemble. The scaled lattice was then allowed to relax to its equilibrium unit cell parameter *a* in the NPT ensemble. Each lattice was simulated for a total of 2.5 ns with a timestep of one femtosecond using a Nosé-Hoover thermostat and a Parrinello-Rahman barostat. The IRMOF-74-V superlattice used in all MD simulations consisted of 16 channels (4 channels × 4 channels).

**Grand-canonical Monte Carlo simulations.** To compute the argon adsorption isotherm in IRMOF-74-V we used Monte Carlo simulations of the grand-canonical ($\mu$VT) ensemble. The DFT-optimized unit cells were used to simulate adsorption isotherms in a standard lattice, while framework coordinates from a snapshot of an MD simulation were used to make adsorption isotherms of a deformed lattice. In these simulations we used as trial moves insertions at a random argon position, deletions, and translations with equal probability using the intermolecular potentials described above. We used the ideal gas law to relate the chemical potential $\mu$ of argon to the pressure in the experiment. Starting at the lowest pressure, we simulated the argon adsorption successively by initiating the simulation with the adsorbate positions stored at the end of the previous simulation. Hysteresis is monitored by performing these GCMC calculations both with increasing and decreasing pressures. We utilized 75,000 equilibrium cycles and 30,000 sampling cycles, where a cycle is defined as max(20, *n*) Markov chain transition proposals, where *n* is the number of adsorbates currently in the system. We replicated the IRMOF-74-V crystal structure three times in the *a* direction (along the axis of the metal helices) to prevent atoms from interacting with their own image.

Our methodology for performing simulations without cross-channel adsorbate interactions is described in detail in Supplementary Figs 3 and 5.

**Study of additional adsorbates and lattices.** We investigated lattice deformation in other systems by imposing a dip in the unit cell parameter rather than relying on NPT simulations. Figure 7a,b shows deformed lattices of IRMOF-74-V loaded with $N_2$ and $CO_2$ gas, respectively, and Fig. 7c shows a deformed lattice of IRMOF-74-V-hex loaded with argon. These systems were also studied by Cho *et al.* We decreased the unit cell parameter of IRMOF-74-V and IRMOF-74-V-hex by 90%, and then increased the unit cell parameter again to the experimentally observed minimum value, at which point we let the system equilibrate in the NVT ensemble. Throughout the simulation, the lattices were loaded with the number of adsorbates commensurate with the isotherms measured by Cho *et al.* $N_2$, $CO_2$ and argon adsorbate systems were simulated at 77, 194 and 87 K, respectively.

Buckingham parameters modelling the interactions of $CO_2$ with the framework were adapted directly from the work of Mercado *et al.*[43], the TraPPE force field was used to model $N_2$–$N_2$ interactions[44] and the EPM2 force field was used to model $CO_2$–$CO_2$ interactions[45]. Non-bonded interactions between $N_2$ and framework atoms were calculated using Lorenz-Berthelot mixing rules with the TraPPE force field for $N_2$ and the CVFF and Duarte *et al.* force field described above for the framework.

Simulations of argon adsorbing in IRMOF-74-III and IRMOF-74-VII used the same force field as argon adsorbing in IRMOF-74-V, but we started with DFT-optimized lattices and atomic charges available in literature[46].

**Calculations of X-ray spectra.** Powder X-ray diffraction patterns of the MOF lattice and SAXS profiles of adsorbing argon are calculated directly from atomic coordinates[47,48]. The experimental samples used by Cho et al. were powder and therefore simulated powder X-ray diffraction peaks can be compared with peaks in the experimental SAXS profile. To create the simulated SAXS profiles shown in Fig. 6b, argon positions are extracted from both a deformed lattice with a loading commensurate with the adsorption isotherm at 0.425 bar and a standard lattice artificially loaded with a number of atoms corresponding to 0.425 bar.

**Calculation of infrared spectra.** The IR spectrum of IRMOF-74 was computed by taking the Fourier transform of the net dipole moment's autocorrelation function obtained from the MD simulation. A quantum correction of $\beta\hbar\omega/(1 - \exp(-\beta\hbar\omega))$ was then applied, as recommended by Gaigeot and Sprik[49].

**Data availability.** CIF files of the DFT-relaxed IRMOF-74-V primitive unit cell (Supplementary Data set 1) and a representative deformed lattice (Supplementary Data set 2) are provided and include REPEAT atomic charges. LAMMPS input files for the study of the deformed IRMOF-74-V lattice with argon present (Supplementary Data sets 3 and 4) are also provided.

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

## Acknowledgements

This research used resources of the National Energy Research Scientific Computing Center, a DOE Office of Science User Facility supported by the Office of Science of the U.S. Department of Energy under Contract No DE-AC02-05CH11231. M.W., C.M.S. and E.B. are supported by the Center for Gas Separations Relevant to Clean Energy Technologies, an Energy Frontier Research Center funded by the U.S. Department of Energy, Office of Science, Basic Energy Sciences under award number DE-SC0001015. D.T. is supported by the National Center of Competence in Research (NCCR) 'Materials Revolution: Computational Design and Discovery of Novel Materials (MARVEL)' of the Swiss National Science Foundation (SNSF) and by the European Research Council (ERC) under the European Union's Horizon 2020 research and innovation programme

(grant agreement No 666983, MaGic). S.J. acknowledges support from an NSF Graduate Research Fellowship.

## Author contributions

All authors planned the research and edited the manuscript. S.J. and C.M.S. executed simulations and E.B. performed DFT calculations, and S.J. also composed the manuscript.

## Additional information

**Competing financial interests:** The authors declare no competing financial interests.

**Publisher's note**: 

