## [Peer Review File · Nature Communications]

Reviewers' comments:

Reviewer #1 (Remarks to the Author):

This is an interesting computational study of argon adsorption in IRMOF-74-V, offering an alternative interpretation to the recent experimental observations of Cho et al. It is sound and the hypothesis tested is attractive, although the authors do not fully back it up with the data gathered and do not justify why it might be a preferable alternative, i.e. they do not offer direct and clear experimental evidence that the Cho interpretation is lacking in merit. There are, in addition, several questions or imprecision (some minor, some more important) that need to be addressed before publication:

- The abstract states that "changes in the crystal structure have so far been ignored as a possibility in the literature". This statement is overly broad, as Cho et al actually did consider and quantify the changes in the structure's unit cell parameter.
- The authors say (in the abstract and the conclusion) that the deformation observed here is "unique in its length scale". It puzzles me for some time what they mean, because the deformation has a well-defined length scale of a few unit cells... while many other materials have deformations that coordinate the entire crystal. So I suggest they might rephrase this to highlight the pattern resulting from it, rather than the length scale itself (which is nothing extraordinary in my view).
- As a related note, one example of such deformation which is both of very large amplitude and of large length scale (because it requires the cooperative motion of linkers across an entire ~ 50 Å unit cell) is that of Kaskel's DUT-49 negative gas adsorption. This might be cited among the notable flexible MOFs given in introduction. It might also be included in the "length scale" discussion (see above).
- In discussing the GCMC results, the authors wave away the "quantitative differences" between experimental and theoretical isotherms. They lay the blame on the rigid lattice model and defects in experimental crystals. However, this seems rather easily dealt with, and not entirely plausible. First, the flexibility of the unit cell does not play a role at low and high loading, where the agreement with experimental data is terrible. Moreover, while defects could explain the saturation loading being too high, they wouldn't at all account for the Henry's constant (low pressure behavior) being off by more than a factor of two. This would rather hint a some problem with the Ar-MOF force field terms, which could be fixed or at least checked (for example, by comparing heats of adsorption with published data).
- GCMC results: can the authors give the values of molar enthalpies for cross-channel Ar-Ar interactions and intra-channel Ar-Ar interactions. This would help quantify a bit the effects that are discussed here.
- GCMC: it is not discussed here how the cross-channel interactions are turned off exactly. If this is done by patching of the LAMMPS software, it would be good practice to include as Supporting Information the patch so others can review it or use it.
- While the authors discuss in detail the nature of the deformed structure obtained upon adsorption and decrease of unit cell parameter, some key elements of basic characterization are not given. The average structure should be provided (e.g. in CIF format), as well as its properties in a Table (unit cell parameters, density, etc.). More importantly, what is its space group and primitive unit cell? This is crucial as it seems that the structural changes observed here are a true phase transition, with lowering of symmetry. So: what is the resultant symmetry? And is the obtained phase metastable, i.e. can it exist in the absence of guest? (DFT calculations could be helpful there to avoid relying on the force field for such a crucial test.)

- The force field put together by the authors and used here is purely validated on structural quantities (unit cell parameters). The fact that it does a good job at this does not mean it would well reproduce dynamical or mechanical properties of IRMOF-74-V. At least some of those should be calculated and checked, because the use of the force field is crucial to the study on flexibility. I suggest looking at vibration modes and bulk modulus, both of which can be calculated both at DFT and force field level and compared.

- "Most studies of flexible MOFs have reported simple changes in lattice parameters": this statement is again much too strong, as evidenced by the very large literature on adsorption-induced and pressure-induced phase transitions in MOFs (including full crystallographic characterization of all phases), done by groups such as Chapman, Moggach, Cheetham, etc.

- The authors qualify the deformation observed upon loading as a "breathing" phenomenon. Breathing is an adsorption-induced first order transition between metastable phases of a material. I do not think this is established here as being the case. It is more an adsorption-induced shrinking at intermediate pore loading, something that has been very widely studied and reported in all manner of microporous materials, from carbons, zeolites, MOFs...

- For the sake of reproducibility of the study, the authors could give some representative input files for their calculations: DFT, GCMC, MD. Also, the CIF structure of the DFT-optimized MOF and its deformed counterpart should be given as supporting information.

Reviewer #2 (Remarks to the Author):

This paper is motivated by a recent work by Cho et al., which suggested that adsorbate-adsorbate interactions across channel walls can drive adsorption in the IRMOF-74 series. From molecular simulations on IRMOF-74-V, the authors found that cross-channel argon interactions have only a minor effect on the adsorption isotherm and do not lead to the argon superlattice described by Cho et al. They argued that argon adsorption induces a new deformation pattern in the crystal lattice of IRMOF-74-V. Although this manuscript describes an interesting scientific discovery, the deformation mechanism has not been fully unveiled because simulations have been performed for only one structure among the IRMOF-74 series. I'm curious if the same deformation mechanism is also observed in other IRMOF-74 materials containing shorter or longer ligands or IRMOF-74-V-hex with side groups. I'm also curious if similar deformations can be induced by the adsorption of other adsorbate molecules in IRMOF-74 series. Some comparison studies on other IRMOF-74 MOFs or other adsorbates will be necessary to further confirm the deformation mechanism. In my opinion, the current version of this work lacks the merit to be published in this high impact journal and will fit another more specialized journal such as Journal of Physical Chemistry Letters.

1. The title of this paper seems to be too generalized. Without the further studies on other IRMOF-74 series or other adsorbates, the title should be changed to a less generalized one. One example may be "Argon-induced lattice deformation in IRMOF-74-V".

2. From Fig. 3, the authors argued that their simulations agree well with the experimentally observed behavior in the lattice parameter. But, some explanations may be required for the discrepancies between experiments and simulations. Why did the simulated unit cell parameter show a more drastic change in the narrow pressure range?

3. Figure 5b: The Ar uptakes just before the step increase at 0.4 bar are higher than those for argon adsorption in a rigid lattice model shown in Figure 2a. Some explanations may be needed.

4. Some confusions can be originated from the name of the MOF, IRMOF-74-V. Some people may guess that this MOF has a vanadium metal in it.

Reviewer #3 (Remarks to the Author):

A. Summary of key results

The authors present a novel observation that of argon adsorption in IRMOF-74 causes channel deformations (rather than argon-density-based superlattices) and that explains the XRD pattern observed in prior experimental measurements. There is a higher level lesson in the paper as well, namely that the MOF community too frequently assumes that MOF crystal structures are rigid.

B. Originality and interest: if not novel, please give references

The results are original and I believe will be of great interest to the MOF community (both computational and experimental).

C. Data & methodology: validity of approach, quality of data, quality of presentation

The results are clearly presented and the use of classical molecular simulations to propose channel deformations in IRMOF-74 are appropriate. Further experimental work will likely be needed to confirm the channel deformation theory presented in this paper, but on its own the evidence in this manuscript is very compelling. It is worth noting that the exact nature/dimensions of the channel deformations is besides the point; the fact that channel deformations can lead to similar XRD patterns as observed and (incorrectly) attributed to gas-density superlattices is a sufficiently important observation.

D. Appropriate use of statistics and treatment of uncertainties

Yes

E. Conclusions: robustness, validity, reliability & F. Suggested improvements: experiments, data for possible revision

My one suggestion that I believe would make the argument more convincing would be to examine system size effects. Clearly a smaller system would not have led to the observed results, so one can't help but wonder what kind of channel deformation patterns would be observed for a system that had 6x6 channels or 8x8 channels instead of just 4x4. Such calculations should not take the authors too much time to perform and include in a revised paper.

The authors observe hysteresis in their GCMC simulations. This has been observed by others but is highly counter-intuitive for a method that ought to be time/path independent. A brief explanation for why hysteresis is observed here, or a references to discussions of GCMC-based hysteresis in adsorption would be greatly appreciated.

G. References: appropriate credit to previous work?

Yes.

H. Clarity and context: lucidity of abstract/summary, appropriateness of abstract, introduction and conclusions

The writing is very clear and easy to follow. The paper was very interesting to read.

Reviewer #1 (Remarks to the Author):

This is an interesting computational study of argon adsorption in IRMOF-74-V, offering an alternative interpretation to the recent experimental observations of Cho et al. It is sound and the hypothesis tested is attractive, although the authors do not fully back it up with the data gathered and do not justify why it might be a preferable alternative, i.e. they do not offer direct and clear experimental evidence that the Cho interpretation is lacking in merit. There are, in addition, several questions or imprecision (some minor, some more important) that need to be addressed before publication:

REPLY: First of all, we would like to make clear that we are **not** disputing the experimental data of Cho et al. We **only** challenge the interpretation of the data. Cho et al. interpreted their data incorrectly assuming that the MOF crystal uniformly expands and shrinks. We also would like to emphasize that their interpretation is not based on any molecular theory or quantitative model, but based on matching distances in the MOF with distances in their X-ray spectra. As their uniformly shrinking MOF cannot explain the spectra, Cho et al. speculated that long-ranged ordering of Ar across channels must cause these additional peaks, which only makes sense if Ar interacts across channels to induce this effect.

Our simulation **quantitatively** predicts all the relevant peaks of the X-ray spectrum of Cho et al. We cannot envision a better confirmation of our simulations than this agreement with the experimental data of Cho et al.

- The abstract states that "changes in the crystal structure have so far been ignored as a possibility in the literature". This statement is overly broad, as Cho et al actually did consider and quantify the changes in the structure's unit cell parameter.

REPLY: We have changed the word in the abstract from "structure" to "symmetry" to emphasize that Cho et al. only considered uniform expansion or shrinking of the unit cell.

- The authors say (in the abstract and the conclusion) that the deformation observed here is "unique in its length scale". It puzzles me for some time what they mean, because the deformation has a well-defined length scale of a few unit cells... while many other materials have deformations that coordinate the entire crystal. So I suggest they might rephrase this to highlight the pattern resulting from it, rather than the length scale itself (which is nothing extraordinary in my view).

REPLY: Indeed this is confusing. We have modified the abstract and conclusion sections to make clear that the deformation occurs over multiple unit cells of the original crystal lattice, and therefore cannot be described using only the atoms of a single unit cell. To our knowledge, such an observation is unique in metal-organic frameworks.

- As a related note, one example of such deformation which is both of very large amplitude and of large length scale (because it requires the cooperative motion of linkers across an entire ~50 Å unit cell) is that of Kaskel's DUT-49 negative gas adsorption. This

might be cited among the notable flexible MOFs given in introduction. It might also be included in the "length scale" discussion (see above).

REPLY: We have included a reference to this work.

- In discussing the GCMC results, the authors wave away the "quantitative differences" between experimental and theoretical isotherms. They lay the blame on the rigid lattice model and defects in experimental crystals. However, this seems rather easily dealt with, and not entirely plausible. First, the flexibility of the unit cell does not play a role at low and high loading, where the agreement with experimental data is terrible. Moreover, while defects could explain the saturation loading being too high, they wouldn't at all account for the Henry's constant (low pressure behavior) being off by more than a factor of two. This would rather hint a some problem with the Ar-MOF force field terms, which could be fixed or at least checked (for example, by comparing heats of adsorption with published data).

REPLY: We agree with the reviewer that the quantitative agreement is not perfect, but we disagree that our force field is inadequate for the purpose of this study. The fact that we overestimate the Henry coefficient by a factor of two and the saturation loading by 20% is a similar agreement with experiment compared with other studies on noble gas adsorption in MOFs. Recently, the Sholl group made an attempt to improve the parameterization of the force fields for Ar and Xe in MOFs based on DFT calculations (see Demir *et al.*, J. Mater. Chem. A, 2015, doi: 10.1039/C5TA06201B). The results showed that DFT force fields were on average of the same quality as off-the-shelf force fields, because DFT functionals could not reliably describe noble gas-MOF interactions. But, the reviewer has a point about the force field and in the revised manuscript we now mention uncertainties in the force field as an additional source for the quantitative differences.

- GCMC results: can the authors give the values of molar enthalpies for cross-channel Ar-Ar interactions and intra-channel Ar-Ar interactions. This would help quantify a bit the effects that are discussed here.

REPLY: We have added the following information with more details to the SI:

We performed *NVT* simulations in an IRMOF-74-V structure with four channels, shown in the SI in Figure SI-5(a), and partitioned the Ar-Ar interaction energy into intra-channel and cross-channel components. Figure 1 shows the ensemble average interaction energy of Ar with (i) other Ar in the same channel and (ii) Ar in different channels. Note the log scale on the y-axis: this shows that the intra-channel interactions are an order of magnitude greater than the cross-channel Ar-Ar interactions.

Figure 1. Partitioning interactions of Ar in the home channel into cross-channel and intra-channel interactions. Shown is the resulting ensemble average interaction energy of Ar in the channel $\langle E_{Ar-Ar} \rangle$ during an NVT simulation with $T = 87$ K, partitioned into intra-channel and cross-channel interactions. The x -axis shows the N (loading) in the NVT simulation. Note that the y -axis is on a log scale.

- GCMC: it is not discussed here how the cross-channel interactions are turned off exactly. If this is done by patching of the LAMMPS software, it would be good practice to include as Supplementary Information the patch so others can review it or use it.

REPLY: We have added this description to the SI.

- While the authors discuss in detail the nature of the deformed structure obtained upon adsorption and decrease of unit cell parameter, some key elements of basic characterization are not given. The average structure should be provided (e.g. in CIF format), as well as its properties in a Table (unit cell parameters, density, etc.). More importantly, what is its space group and primitive unit cell? This is crucial as it seems that the structural changes observed here are a true phase transition, with lowering of symmetry. So: what is the resultant symmetry? And is the obtained phase metastable, i.e. can it exist in the absence of guest? (DFT calculations could be helpful there to avoid relying on the force field for such a crucial test.)

REPLY: We appreciate that the reviewer requests further details. We will address each part of this comment separately.

CIF files of standard and representative deformed lattices were actually included with the first submission, however the journal converted these files into PDFs and distributed them in a merged PDF. We will ensure these files are available in the correct format.

A comparison of DFT and force field predicted unit cell parameters have now been included in the SI.

We agree with the reviewer, the deformation is a lowering of the crystal symmetry (as is clear from the new lattice peaks shown in Figure 6 in the main text). We have previously invested time in this question and, unfortunately, have not been able to answer it. The Spglib library for Python can be used to identify crystal symmetry. The Spglib method relies on atoms being within a given tolerance of their perfectly symmetric position. Our deformed crystal lattice is identified using molecular simulations at 87 K (we explain why this is necessary in the next paragraph). Even at this low temperature, the thermal disorder introduced into the crystal is large enough to cause the Spglib method to fail for our representative structure. Without using a specialized tool, we are not confident that we can identify by inspection all possible symmetry operations on the deformed unit cell. We are however confident that the peaks in our x-ray pattern describe coherent deformation.

We agree with the reviewer that DFT calculations would help assess the metastability of the deformed lattice, however the system size poses problems in this regard. As we described in an earlier answer, the lattice deformation must be modeled using multiple unit cells of the crystal (12 at a minimum). Since a single unit cell of IRMOF-74-V has 228 atoms, and many Ar should be modeled, we have unfortunately far too many atoms to do a geometry relaxation with DFT.

- The force field put together by the authors and used here is purely validated on structural quantities (unit cell parameters). The fact that it does a good job at this does not mean it would well reproduce dynamical or mechanical properties of IRMOF-74-V. At least some of those should be calculated and checked, because the use of the force field is crucial to the study on flexibility. I suggest looking at vibration modes and bulk modulus, both of which can be calculated both at DFT and force field level and compared.

REPLY: We have added this comparison to the SI:

To obtain some more insight into the reliability of the force field, we have calculated an infrared (IR) spectrum for IRMOF-74 (also known as Mg-MOF-74 or Mg₂(dobdc)) and compared our results to the experimental spectrum measured by Tan *et al.* (Chem. Mater., 2014, doi: 10.1021/cm5038183). The comparison is shown below in Figure 2.

In the wavelength regime ($> 500 \text{ cm}^{-1}$) where experimental data is available, there is reasonable agreement between the simulated and experimental spectra. This tells us that the force field parameters describe the motion of the organic linkers reasonably well. Magnesium oxide crystals are known to exhibit absorption around 500 cm^{-1} (see Luxon *et*

al., Phys. Rev., 1969, doi: 10.1103/PhysRev.188.1345). We are therefore pleased to see that the simulated IR spectrum exhibits peaks near 500 cm^{-1} .

Figure 2. Simulated and experimental IR spectra of IRMOF-74 at room temperature. In the wavelength regime ($> 500\text{ cm}^{-1}$) where experimental data is provided, there is reasonable agreement between the simulated and experimental spectra. The simulated peaks near 500 cm^{-1} can be attributed to Mg-O bonds (see Luxon *et al.*, Phys. Rev., 1969, doi: 10.1103/PhysRev.188.1345).

- "Most studies of flexible MOFs have reported simple changes in lattice parameters": this statement is again much too strong, as evidenced by the very large literature on adsorption-induced and pressure-induced phase transitions in MOFs (including full crystallographic characterization of all phases), done by groups such as Chapman, Moggach, Cheetham, etc.

REPLY: We have edited this sentence to more specifically reference studies in literature that report on the expected rigid behavior of IRMOF-74.

- The authors qualify the deformation observed upon loading as a "breathing" phenomenon. Breathing is an adsorption-induced first order transition between metastable phases of a material. I do not think this is established here as being the case. It is more an adsorption-induced shrinking at intermediate pore loading, something that has been very widely studied and reported in all manner of microporous materials, from carbons, zeolites, MOFs...

REPLY: The reviewer is correct to note that we do not characterize the deformed lattice as a metastable state. However, we refer to a Comment published in PCCP (Coudert *et*

al., Phys. Chem. Chem. Phys., 2014, doi: 10.1039/C3CP54042A) on this topic, which drew parallels between the mechanism behind breathing MOFs and other adsorbate-induced volume shrinking. Therefore we find it reasonable to compare the deformation phenomenon we observe to breathing, even if we do not classify IRMOF-74-V as a breathing MOF. But the reviewer does have a point, and we mention these different definitions in the revised manuscript.

- For the sake of reproducibility of the study, the authors could give some representative input files for their calculations: DFT, GCMC, MD. Also, the CIF structure of the DFT-optimized MOF and its deformed counterpart should be given as supporting information.

REPLY: At the submission stage we cannot provide files in all formats, we will work with the editor to ensure that we can provide all relevant files for others to reproduce our work.

Reviewer #2 (Remarks to the Author):

This paper is motivated by a recent work by Cho et al., which suggested that adsorbate-adsorbate interactions across channel walls can drive adsorption in the IRMOF-74 series. From molecular simulations on IRMOF-74-V, the authors found that cross-channel argon interactions have only a minor effect on the adsorption isotherm and do not lead to the argon superlattice described by Cho et al. They argued that argon adsorption induces a new deformation pattern in the crystal lattice of IRMOF-74-V. Although this manuscript describes an interesting scientific discovery, the deformation mechanism has not been fully unveiled because simulations have been performed for only one structure among the IRMOF-74 series. I'm curious if the same deformation mechanism is also observed in other IRMOF-74 materials containing shorter or longer ligands or IRMOF-74-V-hex with side groups. I'm also curious if similar deformations can be induced by the adsorption of other adsorbate molecules in IRMOF-74 series. Some comparison studies on other IRMOF-74 MOFs or other adsorbates will be necessary to further confirm the deformation mechanism. In my opinion, the current version of this work lacks the merit to be published in this high impact journal and will fit another more specialized journal such as Journal of Physical Chemistry Letters.

REPLY: We agree with the reviewer that it is important to provide evidence that our mechanism is more general than the single system we have studied. We therefore have carried out some additional simulations and we are happy to report with different adsorbates and IRMOF-74-V-hex we observed similar deformation. In addition, for IRMOF-74-III and IRMOF-74-VII (without a side chain) we did not observe deformation, indicating a pore size dependence similar to what Cho *et al.* observed. We have included this information in Figure 7 of the main text and the discussion section of the manuscript.

1. The title of this paper seems to be too generalized. Without the further studies on other IRMOF-74 series or other adsorbates, the title should be changed to a less generalized one. One example may be "Argon-induced lattice deformation in IRMOF-74-V".

REPLY: As we noted above, based on the reviewer's comments we have augmented our results to include more data about deformation induced by other adsorbates in other materials in the series.

2. From Fig. 3, the authors argued that their simulations agree well with the experimentally observed behavior in the lattice parameter. But, some explanations may be required for the discrepancies between experiments and simulations. Why did the simulated unit cell parameter show a more drastic change in the narrow pressure range?

REPLY: The distortion is a delicate balance between guest-guest, guest-host and host-host interactions. In light of this, this we were actually surprised to get such a reasonable agreement. We do not have good insight into how uncertainties in these interactions influence our results.

3. *Figure 5b: The Ar uptakes just before the step increase at 0.4 bar are higher than those for argon adsorption in a rigid lattice model shown in Figure 2a. Some explanations may be needed.*

REPLY: We added to the revised manuscript:

The Channel 1 isotherm shown in Fig. 5 and rigid lattice isotherm shown in Fig. 2a are different. The rigid lattice used in Fig. 2a is the DFT-optimized structure, whereas the individual channel isotherms result from molecular simulations and are governed by the force field, which means that the two structures vary and have slightly different sizes and linker orientations. For this reason, Channel 1 should be compared with the deformed channels and lattice rather than the DFT-optimized structure.

4. *Some confusions can be originated from the name of the MOF, IRMOF-74-V. Some people may guess that this MOF has a vanadium metal in it.*

REPLY: We chose to refer to the MOF studied as IRMOF-74-V to maintain consistency with the work of Cho *et al.* To make the composition of the MOF more clear to readers, we listed the components (Mg metal and specific ligand name) when the MOF is introduced in the introduction.

Reviewer #3 (Remarks to the Author):

A. Summary of key results

The authors present a novel observation that of argon adsorption in IRMOF-74 causes channel deformations (rather than argon-density-based superlattices) and that explains the XRD pattern observed in prior experimental measurements. There is a higher level lesson in the paper as well, namely that the MOF community too frequently assumes that MOF crystal structures are rigid.

B. Originality and interest: if not novel, please give references

The results are original and I believe will be of great interest to the MOF community (both computational and experimental).

C. Data & methodology: validity of approach, quality of data, quality of presentation

The results are clearly presented and the use of classical molecular simulations to propose channel deformations in IRMOF-74 are appropriate. Further experimental work will likely be needed to confirm the channel deformation theory presented in this paper, but on its own the evidence in this manuscript is very compelling. It is worth noting that the exact nature/dimensions of the channel deformations is besides the point; the fact that channel deformations can lead to similar XRD patterns as observed and (incorrectly) attributed to gas-density superlattices is a sufficiently important observation.

D. Appropriate use of statistics and treatment of uncertainties

Yes

E. Conclusions: robustness, validity, reliability & F. Suggested improvements: experiments, data for possible revision

My one suggestion that I believe would make the argument more convincing would be to examine system size effects. Clearly a smaller system would not have led to the observed results, so one can't help but wonder what kind of channel deformation patterns would be observed for a system that had 6x6 channels or 8x8 channels instead of just 4x4. Such calculations should not take the authors too much time to perform and include in a revised paper.

REPLY: We have addressed the question regarding the simulation size of the lattice by simulating a 6x6 channel system (36 channels). We mention our result – less perfect deformation – just before the discussion section in the manuscript, and include a representative snapshot along with the resulting X-ray pattern in the SI.

The authors observe hysteresis in their GCMC simulations. This has been observed by others but is highly counter-intuitive for a method that ought to be time/path independent.

A brief explanation for why hysteresis is observed here, or a references to discussions oof GCMC-based hysteresis in adsorption would be greatly appreciated.

REPLY: When we generate desorption curves using GCMC, we carefully reduce the pressure and try to remove adsorbate molecules from a fully loaded structure. Hysteresis arises because the system becomes trapped in metastable states. Over the course of the simulation, the system remains at a higher loading than what is found in the adsorption curve. i.e. the system never has time to fluctuate out of the metastable high-loading state.

In the case of adsorption in a porous material, low- and high-density pores have low free energies in the pressure region around a capillary condensation step, but intermediate densities may have very high free energies that present a difficult barrier which single-particle MC moves cannot easily cross. (see Sarkisov *et al.*, Langmuir, 2000, doi: 10.1021/la001000f)

G. References: appropriate credit to previous work?

Yes.

H. Clarity and context: lucidity of abstract/summary, appropriateness of abstract, introduction and conclusions

The writing is very clear and easy to follow. The paper was very interesting to read.

Reviewers' comments:

Reviewer #1 (Remarks to the Author):

The authors have amended the manuscript to respond to most of the comments. Two have, however, not been addressed:

1. The authors indeed offer an alternative interpretation of the experimental data by Cho et al. While their hypothesis is compatible with experimental data, so is Cho's original interpretation. Thus there is no discriminating factor between the two: both are backed by computational arguments (classical DFT calculations for Cho; force field-based simulations here). But both are compatible with experimental observations. This should be made clearer in the manuscript.

2. The authors argue that the large discrepancies between experimental and theoretical results are inherent to force field approaches. While they cite results from the Sholl group to argue this point, that study shows that there is no *universal* approach that succeeds on all materials: it does not state that force fields tailored to individual materials cannot adequately describe adsorption in that given material.

Moreover, my comment was broader than that: it is not only that the agreement is limited, it has to do with the way it is discussed in the manuscript. Authors lay the blame on the rigid lattice model and defects in experimental crystals, which are factors that can actually be excluded in this specific case: First, the flexibility of the unit cell does not play a role at low and high loading, where the agreement with experimental data is terrible. Moreover, while defects could explain the saturation loading being too high, they wouldn't at all account for the Henry's constant (low pressure behavior) being off by more than a factor of two.

Reviewer #2 (Remarks to the Author):

Now, most of the comments from the reviewers have been answered and revised. However, some of issues still need to be clarified or revised. I recommend the paper for publication if the authors clearly address the following issues.

1. In the revised manuscript, the authors explained the new results in Fig. 7 as follows: "In Fig. 7, we show that the same deformation indeed occurs with other adsorbates and IRMOF-74 analogs. Fig. 7 demonstrates that the adsorbate identity affects the extent of deformation." More discussions may be needed for the new results (e.g. some quantitative analyses on the effect of adsorbate identity on the extent of deformation).

2. "We also modeled argon adsorbing in IRMOF-74-III and IRMOF-74-VII, but did not access deformation in these structures.": The simulation results on argon adsorption in IRMOF-74-III and IRMOF-74-VII should be shown at least in the supporting information.

Reviewer #3 (Remarks to the Author):

The reviewers have adequately addressed my concerns, and the concerns of the other reviewers.

Reviewer #1 (Remarks to the Author):

The authors have amended the manuscript to respond to most of the comments. Two have, however, not been addressed:

1. The authors indeed offer an alternative interpretation of the experimental data by Cho et al. While their hypothesis is compatible with experimental data, so is Cho's original interpretation. Thus there is no discriminating factor between the two: both are backed by computational arguments (classical DFT calculations for Cho; force field-based simulations here). But both are compatible with experimental observations. This should be made clearer in the manuscript.

REPLY: We agree with the reviewer that only additional experiments can unambiguously prove which of the two hypotheses, ours and Cho *et al.*'s, is correct. To make this point more clear, we have modified the first paragraph of the discussion section with:

“The deformed framework reproduces key features that were experimentally observed during argon adsorption by Cho *et al.*, namely changes in the lattice parameter and X-ray pattern. We therefore propose the deformation as an alternative hypothesis to a superlattice of different argon densities in neighboring channels. As both interpretations can explain the additional X-ray peaks, it would therefore be interesting to see whether additional experiments can confirm our proposed deformation.”

*2. The authors argue that the large discrepancies between experimental and theoretical results are inherent to force field approaches. While they cite results from the Sholl group to argue this point, that study shows that there is no *universal* approach that succeeds on all materials: it does not state that force fields tailored to individual materials cannot adequately describe adsorption in that given material.*

Moreover, my comment was broader than that: it is not only that the agreement is limited, it has to do with the way it is discussed in the manuscript. Authors lay the blame on the rigid lattice model and defects in experimental crystals, which are factors that can actually be excluded in this specific case: First, the flexibility of the unit cell does not play a role at low and high loading, where the agreement with experimental data is terrible. Moreover, while defects could explain the saturation loading being too high, they wouldn't at all account for the Henry's constant (low pressure behavior) being off by more than a factor of two.

REPLY: We agree that the discrepancy between simulated and experimental isotherms at low loadings is due to inaccuracies in the force field. We have edited the section of the manuscript where we discuss this as follows:

“The large simulated Henry coefficient, apparent in the overestimation of loading at low pressures, can be attributed to inaccuracies in the force field description of noble gas-MOF interactions, and is in similar agreement compared with other studies on noble gas adsorption in MOFs (see Demir *et al.*, J. Mater. Chem. A, 2015, doi:

10.1039/C5TA06201B). At intermediate and high loadings, the quantitative differences between the experimental and simulated isotherms may be related to the fact that we use a rigid lattice model for all simulations and assume a perfect crystal structure, which is experimentally impossible to synthesize.”

Reviewer #2 (Remarks to the Author):

Now, most of the comments from the reviewers have been answered and revised. However, some of issues still need to be clarified or revised. I recommend the paper for publication if the authors clearly address the following issues.

1. In the revised manuscript, the authors explained the new results in Fig. 7 as follows: “In Fig. 7, we show that the same deformation indeed occurs with other adsorbates and IRMOF-74 analogs. Fig. 7 demonstrates that the adsorbate identity affects the extent of deformation.” More discussions may be needed for the new results (e.g. some quantitative analyses on the effect of adsorbate identity on the extent of deformation).

2. “We also modeled argon adsorbing in IRMOF-74-III and IRMOF-74-VII, but did not access deformation in these structures.”: The simulation results on argon adsorption in IRMOF-74-III and IRMOF-74-VII should be shown at least in the supporting information.

REPLY (to point 1 and 2): We added two new sections to the SI in which we discuss the deformation induced by different gasses in more detail (section 6) and we show the argon adsorption results of IRMOF-74-III and IRMOF-74-VII (section 7).

Reviewer #3 (Remarks to the Author):

The reviewers have adequately addressed my concerns, and the concerns of the other reviewers.